# Substrate Specificity of Biofilms Proximate to Historic Shipwrecks

**DOI:** 10.3390/microorganisms11102416

**Published:** 2023-09-27

**Authors:** Rachel L. Mugge, Rachel D. Moseley, Leila J. Hamdan

**Affiliations:** 1U.S. Naval Research Laboratory, Ocean Sciences Division, Stennis Space Center, MS 39529, USA; rachel.mugge.ctr@nrlssc.navy.mil; 2School of Ocean Science and Engineering, University of Southern Mississippi, Ocean Springs, MS 39564, USA

**Keywords:** biofilms, metagenomics, historic shipwrecks, deep sea, built environment, Gulf of Mexico, wood, steel

## Abstract

The number of built structures on the seabed, such as shipwrecks, energy platforms, and pipelines, is increasing in coastal and offshore regions. These structures, typically composed of steel or wood, are substrates for microbial attachment and biofilm formation. The success of biofilm growth depends on substrate characteristics and local environmental conditions, though it is unclear which feature is dominant in shaping biofilm microbiomes. The goal of this study was to understand the substrate- and site-specific impacts of built structures on short-term biofilm composition and functional potential. Seafloor experiments were conducted wherein steel and wood surfaces were deployed for four months at distances extending up to 115 m away from three historic (>50 years old) shipwrecks in the Gulf of Mexico. DNA from biofilms on the steel and wood was extracted, and metagenomes were sequenced on an Illumina NextSeq. A bioinformatics analysis revealed that the taxonomic composition was significantly different between substrates and sites, with substrate being the primary determining factor. Regardless of site, the steel biofilms had a higher abundance of genes related to biofilm formation, and sulfur, iron, and nitrogen cycling, while the wood biofilms showed a higher abundance of manganese cycling and methanol oxidation genes. This study demonstrates how substrate composition shapes biofilm microbiomes and suggests that marine biofilms may contribute to nutrient cycling at depth. Analyzing the marine biofilm microbiome provides insight into the ecological impact of anthropogenic structures on the seabed.

## 1. Introduction

Upon entering the marine environment, hard surfaces are colonized by microorganisms [1]; this process is initiated by the formation of a conditioning film, which occurs through the adsorption of organic molecules and other dissolved substances from the water column onto the substrate [2]. In the early stages of development, pioneer taxa recruiting to a biofilm can alter the surface properties to achieve a preferable habitat [1,2,3]. Following the establishment of a microbial biofilm, the surface is then suitable for the adhesion of other organisms, such as microalgae [4] and marine diatoms [5]. Biofilms and their associated bacteria can also act as mediators of the larval settlement of invertebrates and macrofauna [6].

For microorganisms, a surface-attached lifestyle offers several advantages over a planktonic lifestyle in the water column. Non-polar organic compounds attracted to surfaces contain higher carbon and nutrient concentrations than the surrounding water, enhancing microbial colonization and activity [7]. Within the exopolysaccharide (EPS) matrix, a key chemical moiety produced by biofilm-forming microorganisms, sessile prokaryotes are protected from predators, toxins, viruses, and metal ions [2,8]. Metabolic cooperation and cell–cell communication can enable the physiological homeostasis of cells within microenvironments in the biofilm [9] while providing resistance to physical changes in the local environment, such as shear stress and ultraviolet radiation [10]. 

The success of microbial attachment and subsequent biofilm formation is dependent on the characteristics of the substrate, including the surface free energy, hydrophobicity, roughness, architecture, and chemical composition [11,12,13,14]. Substrate viscosity and surface tension are determining factors for communities of biofouling microorganisms [15,16], while increased surface roughness and hydrophobicity lead to stronger biofilm attachment [5]. The initial communities that recruit to the biofilm are also influenced by local environmental conditions, including light, temperature, and water depth [12,17]. However, it remains unclear whether substrate characteristics or environmental conditions have a greater impact on community composition [11,14].

Marine biofilm research has centered on biofilm communities on steel and wood, both common anthropogenic materials in the marine environment. Mild steel is frequently used in the construction of ship hulls and marine infrastructure, including docks, moorings, pipelines, and oil platforms [18,19], while wood is a common construction material used in nearshore environments. To better understand microbially influenced corrosion, an unavoidable corrosive process on steel in the marine environment [20], the composition of engineering alloys has been observed to recruit different dominant microorganisms within the biofilm [21]. Recent studies on short-term marine biofilm succession on steel have revealed a prevalence of Rhodobacteraceae; members of the Alpha-, Delta-, and Gammaproteobacteria classes; iron-oxidizing bacteria; and sulfide-oxidizing bacteria [22,23,24,25]. Natural and anthropogenic sources of wood, such as wood falls and wooden shipwrecks, respectively, are also commonly found on the seabed, including at deeper water depths [26]. Wood is an important source of organic matter for wood-degrading microbiomes, from coastal regions to the deep ocean [27]. Wood type has been observed to shape the biofilm microbiome, typically composed of Actinobacteria and Alpha- and Deltaproteobacteria [28,29,30]. 

Pelagic microorganisms are known to play a role in carbon and nutrient cycling in the water column [31]. Particle-attached microorganisms, including those on detritus and marine snow, typically have higher cell densities than pelagic microbiomes, and they have also been recognized as hotspots of carbon cycling [32,33,34]. While it is realized that surface-associated marine microbiomes contribute to various biogeochemical cycles, such as nitrogen fixation [35], sulfate reduction [36], and iron cycling [37], the metabolic contribution of biofilms on built structures to marine ecosystem function is underexplored. As the marine built environment expands [38,39], knowledge about how anthropogenic structures impact the seabed would be improved by examining biofilm metabolisms on common substrates found in the marine environment.

Shipwrecks represent one common type of built structure in the ocean that transform into artificial reef ecosystems. These abundant and ecologically important structures are capable of supporting diverse biological communities within their immediate environment [30,40]. Thus, this study focused on marine biofilm samples collected in areas surrounding two historic wooden-hulled shipwrecks and one historic steel-hulled shipwreck. By examining the metagenomes of biofilms on steel and wood substrates at different locations through a comparative framework, this study aimed to understand the substrate- and site-specific impacts of marine built structures on short-term (3–4 month) biofilm microbiome formation. These data provide new knowledge on how biofilm community composition and functional potential are influenced by the presence of wood and steel on the seabed.

## 2. Methods

### 2.1. Field Experiments

Three historic shipwrecks in the northern Gulf of Mexico were selected as sites for this study. The shipwreck *Anona* is a steel-hulled steam yacht sunk in 1944 in approximately 1250 m of water, described elsewhere [40]. Site 15711 and Site 15470 are two 19th century wooden-hulled sailing ships resting in 525 m and 1800 m water depths, respectively, and they are separated horizontally by 60 nautical miles [41]. Site maps and descriptions of these recently discovered wooden-hulled shipwrecks have been published in prior works [30,41,42]. Microbial Recruitment Experiment (MRE) arrays containing replicate mild carbon steel (C1020) coupons and/or replicate wood (pine, oak) coupons were deployed proximate to each shipwreck, with each successive MRE in a single array deployed at distances extending away from the structure in 25 m increments, up to 115 m (substrates, sites, and exact distances are detailed in Appendix A). In the experimental designs from prior studies [30,43], carbon steel coupons were deployed at *Anona*, while carbon steel, pine, and oak coupons were deployed at Site 15711 and Site 15470. Steel coupon assembly followed an established method [44]. Wood coupon assembly, MRE and lander array design, and lander array deployment and recovery have been detailed previously [30]. MRE lander arrays were deployed from Research Vessel *Point Sur* in June–July 2019 and recovered in November 2019. Immediately following recovery, the wood coupon surfaces were scraped with a flame-sterilized spatula to collect biofilms into Lysing Matrix E tubes containing MT and sodium phosphate buffers (MP Biomedicals), and the steel coupons were aseptically sampled into containers. The samples were stored at −20 °C shipboard, transported to the lab, and stored at −80 °C until DNA extraction. 

### 2.2. DNA Extraction and Metagenome Sequencing

Prior to DNA extraction, the steel coupons were thawed, and biofilms were scraped into Lysing Matrix E tubes using a flame-sterilized spatula. Each tube contained both buffers as above in addition to 10 µL of bovine serum albumin (BSA) to prevent the competitive binding of iron to DNA [45]. Genomic DNA was extracted from all samples following the manufacturer’s protocol of the BIO 101 FastDNA Spin Kit (MP Biomedicals, LLC, Santa Ana, CA, USA) with a few modifications as described in a prior work [46]. Extracted DNA (ng/mL) was quantified on a Qubit 3.0 Fluorometric Quantitation system (Invitrogen, Carlsbad, CA, USA) and checked for purity on a NanoDrop 2000 spectrophotometer (ThermoFisher, Waltham, MA, USA). Exactly 20 µL of each sample was plated on a 96-well plate for metagenome sequencing. 

Metagenomes were sequenced for this study from the down-selection of samples based on the results of the 16S rRNA gene amplicon sequence analysis published in Moseley et al. [30] for wood biofilms and Mugge et al. [43] for steel biofilms. By combining these two similar yet separate experimental designs, a total of 110 biofilm samples were down-selected to 24 metagenome samples, which included steel biofilms at *Anona*; steel, pine, and oak biofilms at Site 15711; and pine and oak biofilms at Site 15470 (Appendix A). These samples were statistically representative of the dataset and were selected based on sequence counts, Shannon alpha diversity, and taxonomic composition from 16S rRNA gene sequencing data. Metagenome sequences were generated at the Integrated Microbiome Resource (IMR) at Dalhousie University (Halifax, NS, Canada). An Illumina Nextera Flex Library Preparation Kit was used to prepare metagenome libraries from a starting amount of 1 ng per sample [47]. Following PCR purification and normalization using the Just-a-Plate 96 Kit (Charm Biotech, San Diego, CA, USA), sequencing was performed on an Illumina NextSeq 550 System to generate 150 bp paired-end sequences. 

### 2.3. Bioinformatics

Metagenome sequence pre-processing followed Metagenomics Standard Operating Procedure v3 available in the Microbiome Helper repository [47]. To begin, multiple lanes of sequencing were concatenated together per sample, and the read quality was inspected using FastQC [48]. Paired-end sequences were then processed using KneadData [49] to trim sequences, using Trimmomatic to remove low-quality sequences [50], and then using BowTie2 to screen out contaminant reads mapping to human and PhiX genomes [51]. GNU Parallel [52] was used to process multiple samples through KneadData in parallel. A downstream analysis followed select steps from the Assembly and Metagenome Assembled Genomes Tutorial publicly available on GitHub [53]. Quality controlled paired-end sequences were assembled into contiguous sequences (contigs) using MetaSpades [54], and assemblies were evaluated using MetaQuast [55] to obtain per-sample assembly statistics. Pullseq [56] was used to filter assemblies to 500 bp contigs and longer, and then BowTie2 was used to map quality controlled, paired-end short reads to contigs to obtain the assembly percent coverage within each sample.

### 2.4. Metagenome Taxonomic and Functional Annotation

Taxonomic annotation was performed on assembled contigs using Kaiju [57] with default parameters to query sequences against NCBI’s nonredundant (nr) protein +euk reference database [58]. The ‘kaiju2table’ command was used to create relative abundance summary tables from the Kaiju output of individual samples. Summary tables included a table of all taxa at the species level as input for a statistical analysis, a table filtered to exclude <1% of taxa at the species level to understand dominant organisms, and a table filtered to exclude <0.1% of taxa at the class level to obtain the taxonomic profiles of each metagenome. Shannon alpha diversity, observed features (richness), and Pielou’s evenness were calculated on the relative abundance table at the species level within QIIME2 version 2022.2 [59]. Two-sample *t*-tests assuming equal variance were run in Excel on the Shannon diversity data to discover statistically significant differences in diversity between the same substrates at different sites (steel at *Anona* vs. steel at Site 15711; pine at Site 15711 vs. pine at Site 15470; oak at Site 15711 vs. oak at Site 15470). Filtered class tables were imported into R/RStudio (v. 4.1) and merged using the ‘dplyr’ package [60]. The merged class table was manually annotated by phylum and visualized as a bubble plot using ggplot2 [61].

To infer the genomic potential within the steel and wood biofilm metagenomes, targeted functional annotation was performed using MagicLamp [62], a platform that uses curated Hidden Markov Models (HMMs) to search for and annotate functional genes of interest. Within MagicLamp, the LithoGenie pipeline was used to discover genes related to microbial chemolithotrophic metabolisms, and the WspGenie pipeline was used to discover genes related to biofilm formation functions. For each pipeline, Prodigal [63] was used to predict open reading frames (ORFs) on filtered contigs within each metagenome. Then, gene annotation was performed using hmmsearch within HMMER [64] to query a custom library of HMM sets against predicted ORFs. Finally, the ‘--norm’ flag was used to normalize annotated genes by ORFs on each contig, and an output table summarized the normalized gene abundance across all metagenomes. In LithoGenie, normalized abundance tables were created for metabolic genes related to specific elements (i.e., carbon). The final output tables were visualized in R/RStudio as described above for taxonomy. 

Profile HMMs within LithoGenie and WspGenie were sourced from Pfam [65], TIGRFAMs [66], Anantharaman et al. [67], and custom HMMs created and curated following the method detailed in Garber et al. [68]. For custom HMMs, representative protein sequences were obtained from UniProtKB [69], and bitscore cutoffs were calibrated against the NCBI nonredundant (nr) protein reference database [58].

### 2.5. Statistical Analysis

A statistical analysis was performed using version 7 of the Plymouth Routines in Multivariate Ecological Research (PRIMER) plus PERMANOVA software (PRIMER-E Ltd., Plymouth, UK) and followed select steps for an analysis of multivariate data [70]. For taxonomy, a Bray–Curtis dissimilarity matrix was created from the merged relative abundance table at the species level. Hierarchical clustering (CLUSTER) was run to generate a similarity dendrogram, which was then overlaid on a two-dimensional non-metric multidimensional scaling (NMDS) plot to visualize differences in community composition. To understand statistically significant differences between the sites, substrates, distances from structures, and interactions of these factors, a permutational multivariate analysis of variance (PERMANOVA) main test using a full model (factors included ‘Distance’, ‘Substrate’, ‘Site’, ‘Distance × Substrate’, ‘Distance × Site’, ‘Substrate × Site’) was run on the taxonomic Bray–Curtis dissimilarity matrix using distance from structure as a covariate, site and substrate as fixed factors, Type I sum of squares, fixed effects sum to zero to mixed terms, permutation of residuals under a reduced model, and 999 permutations. Each factor and interaction term were examined for its contribution to the model by considering the *p*-value, pseudo-F statistic, and estimated components of variation. Using a cutoff *p*-value according to the methods of [71,72], factors and interaction terms were removed from the model if the corresponding *p*-value was greater than 0.25 [70], and the PERMANOVA was re-run using this reduced model format. *p*-values in the text are reported from reduced models, except for in cases where a reduced model was not required (i.e., where all terms except for one have a *p*-value greater than 0.25). Additionally, each full and reduced model was run using a Type I (sequential) sum of squares, except for in cases where the reduced model did not contain the covariate factor (distance from structure) or any interaction terms with the covariate; in this case, a Type III (partial) sum of squares was used within the model. If the main PERMANOVA test revealed a significant interaction term between the fixed factors of interest (substrate, site), a pairwise PERMANOVA was run on each factor within the interaction term; for non-significant interaction terms, pairwise tests were run only when individual factors were significant.

For a statistical analysis of metagenome functions, normalized gene abundance tables were used as input to PRIMER7, including a gene abundance table of biofilm formation functions and gene abundance tables for each of the ten elements. Bray–Curtis dissimilarity matrices were created for each abundance table, and a hierarchical clustering dendrogram and an NMDS plot were created only for biofilm formation functions. Using parameters and the method described above, PERMANOVA main tests were run on each dissimilarity matrix to discover statistically significant differences between the substrate and site, considering the effects of distance from structure, within each category of metagenomic function.

## 3. Results

### 3.1. Quality and Descriptive Statistics of Assembled Metagenomes

Oak biofilms had the highest number of raw, paired-end reads (average of 4,240,336) compared to pine biofilms (average of 3,540,881) and steel biofilms (average of 3,028,041) (Appendix A). Of these, an average of 83–89% of reads were retained following sequence quality control across all samples. Following the assembly of short-sequence reads into longer, contiguous sequences (contigs), assembly statistics revealed that the oak biofilms had the highest number of contigs greater than 500 base pairs (average of 56,639) compared to the pine biofilms (average of 49,726) and steel biofilms (average of 15,455). However, the average N50 values, a measure of the contiguity of an assembly [73], were the highest in the steel biofilms (average of 2471), followed by the oak biofilms (average of 1531) and pine biofilms (average of 927). The mapping rates of reads to assemblies were the highest in the oak (average of 60.22% mapped) and steel (average of 59.59% mapped) biofilms and the lowest in the pine biofilms (average of 33.22% mapped).

### 3.2. Diversity, Richness, and Evenness of Biofilm Metagenomes

The Shannon alpha diversity was higher in the oak (average of 9.96) and pine (average of 9.25) biofilms than in the steel biofilms (average of 7.12) (Appendix A). The oak biofilms had the same average Shannon diversity at Sites 15711 and 15470 (10.21), while the pine biofilms had a higher average diversity at Site 15711 (9.57) than at Site 15470 (8.92). The steel biofilm diversity was significantly different between *Anona* and Site 15711 (*p* = 0.002) (Appendix A), while the pine biofilm diversity was not significantly different between Sites 15711 and 15470 (*p* = 0.066) (Appendix A). The oak biofilm diversity was also not significantly different between Sites 15711 and 15470 (*p* = 0.071) (Appendix A). The steel biofilms had the lowest richness (average of 2052 observed features) and evenness (average of 0.65) compared to the pine biofilms, which had a greater richness (average of 5388 observed features) and evenness (average of 0.75), with the oak biofilms having the highest richness (average of 6123 observed features) and evenness (average of 0.79). 

### 3.3. Biofilm Taxonomic Composition

The steel biofilm communities had the highest percentage of classified contigs (average of 81.34%) compared to the pine (average of 66.83%) and oak (average of 69.94%) biofilm communities (Appendix A). Regardless of substrate, site, or distance from structure, the composition of all biofilms was dominated by Gammaproteobacteria (steel, average of 31.5%; pine, average of 12.0%; oak, average of 21.0%) and Alphaproteobacteria (steel, average of 10.8%; pine, average of 15.7%; oak, average of 18.4%) (Figure 1). The bacterial classes present in lesser abundance across all biofilms included Betaproteobacteria (steel, average of 3.02%; pine, average of 3.3%; oak, average of 3.7%) and Deltaproteobacteria (steel, average of 0.7%; pine, average of 3.7%; oak, average of 4.2%), although Deltaproteobacteria were overall more abundant in the wood biofilms than in the steel biofilms. Other classes contributing to community abundance in the steel biofilms were Epsilonproteobacteria (average of 19.5%) and Zetaproteobacteria (average of 6.6%). Two classes from the Bacteroidetes phylum that were important contributors to the wood biofilm communities were Flavobacteriia (pine, average of 14.2%; oak, average of 11.7%) and Cytophagia (pine, average of 2.3%; oak, average of 1.1%). Within the Verrucomicrobia phylum, Opitutae were unique to the pine biofilm communities (average of 2.8%), while Verrucomicrobiae were present in all wood biofilms, with a higher abundance on the pine (average of 1.6%) than on the oak (average of 0.02%). 

At the highest resolution of taxonomic classification, the biofilm communities clustered primarily by substrate (steel, pine, and oak) (Figure 2). *Anona* and Site 15,711 formed separate clusters for the steel samples, while the pine and oak samples clustered together at both Site 15711 and Site 15470. However, the hierarchical cluster analysis revealed site-specific clusters of samples under their larger respective substrate clusters. Following the results from the PERMANOVA main test of the full model (Appendix A), the reduced model revealed that the factors ‘substrate’ (*p* = 0.001), ‘site’ (*p* = 0.001), and ‘distance from structure’ (*p* = 0.031) were all significant factors shaping the biofilm taxonomic composition, with a significant interaction between substrate and site (Appendix A). Pairwise PERMANOVA tests for the interaction term showed that the biofilm communities on the steel were statistically different between *Anona* and Site 15711 (*p* = 0.035), the biofilm communities on the pine were statistically different between Site 15711 and Site 15470 (*p* = 0.014), and the biofilm communities on the oak were statistically different between Site 15711 and Site 15470 (*p* = 0.026) (Appendix A). 

At the species level, the steel substrates at *Anona* and Site 15,711 were dominated by a Gammaproteobacteria bacterium (*Anona*, average of 13.7%; Site 15,711, average of 6.5%), *Sulfurimonas* (*Anona*, average of 8.3%; Site 15,711, average of 5.4%), and *Arcobacter* (*Anona*, average of 8.5%; Site 15711, average of 1.4%) (Appendix A). Other dominant taxa on the steel substrates at *Anona* included *Ghiorsea bivora* (average of 3.9%), a Zetaproteobacteria bacterium (average of 1.5%), a Thiotrichaceae bacterium (average of 1.4%), and a Rhizobiales bacterium (average of 1.3%), while other dominant steel taxa at Site 15711 were *Kangiella* (average of 2.8%) and *Cocleimonas flava* (average of 1.1%). A Rhizobiales bacterium also dominated the biofilm communities on the pine substrates at Site 15711 (average of 5.4%) and Site 15470 (average of 6.7%), in addition to a Methylophilaceae bacterium at Site 15711 (average of 1.8%) and at Site 15470 (average of 0.6%) (Appendix A). On the pine at Site 15470, other taxa contributing to the community composition were a Flavobacteriaceae bacterium (average of 4.1%), a SAR86 cluster (Gammaproteobacteria) bacterium (average of 2.2%), and a Verrucomicrobiae bacterium DG1235 (average of 1.9%). The oak biofilms also contained an Methylophilaceae bacterium (Site 15711, average of 1.2%; Site 15470, average of 1.2%) and a SAR86 cluster bacterium (Site 15711, average of 1.6%; Site 15470, average of 1.1%) at both wooden shipwreck sites. At Site 15470, the oak biofilms had a higher diversity of other dominant taxa, including a Rhizobiales bacterium (average of 1.8%), a Planctomycetes bacterium Poly30 (average of 1.5%), and a Rhodobacterales bacterium 56_14_T64 (average of 1.2%).

### 3.4. Biofilm Formation Genes

An NMDS plot and hierarchical clustering dendrogram based on the abundances of biofilm formation genes (identified using WspGenie in MagicLamp) revealed differences primarily between substrates (Figure 3 and Appendix A). While the PERMANOVA (main test, reduced model) showed substrate (*p* = 0.001), distance from structure (*p* = 0.004), and the interaction term ‘distance x substrate’ (*p* = 0.013) to be significant factors impacting biofilm formation gene abundance, site was not a significant factor (*p* = 0.107), and the interaction term ‘substrate x site’ was also not a significant factor (*p* = 0.085) (Appendix A). Pairwise PERMANOVA (reduced model) tests revealed significant differences in biofilm formation gene abundance between the steel and pine (*p* = 0.001), between the steel and oak (*p* = 0.002), and between the pine and oak (*p* = 0.024) (Appendix A). Overall, seven protein domains coding for six genes related to biofilm formation in bacteria (*Wsp* genes) were detected within the dataset, including three genes related to chemotaxis: *WspC*, *WspA*, and *WspB-WspD-WspE* (*CheW*); two genes related to two-component signaling systems: *WspE-WspR* and *WspE*; and one gene related to surface association: *WspR* (Figure 4). The steel biofilms at *Anona* and Site 15711 had an overall higher abundance of biofilm formation genes. Within the steel biofilms, most of these genes generally decreased in abundance with increasing distance at *Anona*, except for *WspR* and *WspA*, which decreased from 29 m to 83 m and then increased in abundance at the farthest distance, 108 m. A similar trend was observed within the steel biofilms at Site 15711, where biofilm formation genes generally decreased from 40 m to 90 m and then increased in abundance at the farthest distance, 115 m. Although the pine and oak biofilms overall had lower abundances of biofilm formation genes, the opposite trend across distance was observed at both Sites 15711 and 15470, where biofilm formation gene abundance generally increased with increasing distance. This pattern held true for the pine biofilms, except for *WspR* at both Sites 15711 and 15470, and *CheW* and *WspE* at Site 15470. In the oak biofilms at Site 15470, *WspA*, *WspE*, and *WspE-WspR* increased in abundance with increasing distance, while *CheW* increased from 22 m to 72 m and then decreased at 97 m.

### 3.5. Chemolithotrophic Metabolisms

Target genes associated with 10 elements involved in microbial chemolithotrophic metabolisms, namely, carbon, nitrogen, oxygen, sulfur, iron, manganese, urea, arsenic, hydrogen, and C1 compounds, were found in this dataset (Appendix A and Figure 5). The PERMANOVA (main test, reduced model) results indicated a high statistical significance of substrate (*p* = 0.001–0.002) for all 10 elements, while site was only statistically significant for genes related to arsenic (*p* = 0.009), carbon (*p* = 0.01), iron (*p* = 0.002), sulfur (*p* = 0.002), and urea (*p* = 0.001) (Appendix A). While distance from structure was not significant for any element, the interaction term ‘substrate x site’ was significant only for arsenic (*p* = 0.004). The steel biofilms had a higher average abundance of genes related to thiosulfate oxidation (*soxY, soxZ, soxX*, and *soxA*), sulfur oxidation (*dsrC, dsrB*, and *dsrA*) and sulfide oxidation (*sqr*), oxygen reduction (*ccoP, ccoP*, and *ccoN*), nitrite reduction (*nirD, nirB*), nitric oxide reduction (*norC* and *norB*), and nitrate reduction (*napB, napA*, and *nasA*). Additionally, iron oxidation (Cyc2 rep Clusters 1 and 3) and iron reduction (*EetA, EetB, DmkA*, and t4ap) genes, carbon fixation (rubisco form I and II CBB, *aclA*, and *aclB*) genes, arsenate reduction (*arsC*, arsenate reductase thioredoxin) genes, and hydrogen oxidation (hydrogenase groups 1, 2a, 2b, 3d, and 4) genes were also higher within the steel biofilms. The wood biofilms had a higher average abundance of genes related to manganese oxidation (*mopA, mnxG*, and *mcoA*), urea hydrolysis (*ureA*, *ureB*, and *ureC*), and C1 cycling, specifically methanol oxidation (*pqq, xoxF*, and *mxaF*), although these abundances were higher in the oak biofilms than in the pine biofilms. 

Site-specific differences in metabolisms included a higher abundance of arsenate reduction genes at *Anona*, the highest abundance of methanol oxidation genes at Site 15470, and the highest abundance of a manganese oxidation gene at Site 15711. Sites 15711 and 15470 had higher abundances of urea hydrolysis genes, while *Anona* had the highest average abundances of iron cycling genes. While Site 15711 had the highest average abundances of sulfur cycling genes, the steel biofilms had the highest average abundance compared to the pine and oak biofilms at this site. A summary of target genes within this dataset, including associated elements, predicted functions and pathways, and HMM sources, can be found in Appendix A.

## 4. Discussion

The goal of this study was to compare the community composition and functional potential of biofilms recruited to different substrates commonly found on the seafloor as a result of human activities. In this work, steel, pine, and oak substrates were placed near three extant historic shipwrecks on the seabed to understand how biofilm recruitment surfaces (i.e., different substrates) impact formation and metabolic potential. Additionally, comparing biofilm microbiomes within the context of substrate availability in the local environment provides insight into the similarities and differences of biofilm functional potential during formation.

The differences in diversity between the steel and wood (pine and oak) biofilms concur with previous results of a higher diversity in wood biofilms analyzed from 16S rRNA gene sequencing data [30] compared to a lower diversity in steel biofilms [74]. The significant differences in the steel biofilm diversity between sites are likely attributable to characteristics of the environment, namely, water depth; Site 15711 rests in 525 m water depth, while *Anona* was sunk in 1250 m of water. These site-specific differences between steel biofilms have been observed previously [43]. Although the pine and oak biofilms sampled from the same sites came from the same seafloor experiments (MREs), the significance of the pine biofilm diversity and the non-significance of the oak biofilm diversity between Sites 15711 and 15470 may be attributed to differences in the wood type [27], such as material properties, one of the determining features of biofilm formation on a substrate [75]. 

Alphaproteobacteria and Gammaproteobacteria are bacterial classes within the Proteobacteria phylum that are typically dominant in developing marine biofilms on plastic, glass, steel, and wood, regardless of laboratory or in situ environment [3,30,44,45,76,77]. Specifically, Alphaproteobacteria, Rhodobacteraceae, and Rhizobiales were observed in the biofilm communities on all three substrates in this study and were likely associated with the abundance of *CheW* and *WspA*, two biofilm formation genes found in all biofilms. These Alphaproteobacteria taxa have also been shown to contribute to biofilm formation in prior studies [78,79,80]. Although the dominant Gammaproteobacteria on the steel substrates was not classified beyond the class level, this class is also frequently found within marine biofilms [45,81], can contribute to short-term biofilm succession on steel [42], and has recently been implicated in the formation of the biofilm matrix [82]. One of the dominant Gammaproteobacterial phylotypes, *Kangiella*, found only in the steel biofilms at Site 15711, is associated with polysaccharide degradation and carbohydrate metabolism [76] and extracellular protein degradation [83]. Multiple dominant phylotypes of the iron-oxidizing class Zetaproteobacteria, and the sulfur-oxidizing *Sulfurimonas* and *Arcobacter*, both within class Epsilonproteobacteria, likely contributed to the greater abundance of iron and sulfur cycling genes, respectively, on the steel surfaces, which contributed to significant metabolic differences between substrates. 

One of the taxa driving community differences between the steel and wood biofilms, with a higher abundance in the wood and oak biofilms, was Flavobacteriia, which was also observed in the 16S rRNA gene sequencing data of these biofilms in a prior study [30]. Regardless of wood type, Flavobacteriia have been shown to be important colonizers of wood surfaces in the marine environment [84] and to be involved in organic matter degradation along with Cytophagia [85], which was also more abundant on the pine and oak surfaces in this study, particularly at Site 15711. Deltaproteobacteria, common wood colonizers in the marine environment, were also important within the pine and oak biofilms, and their abundance has been observed to increase with immersion time [76,86]. The abundance of Betaproteobacteria in the wood samples was mostly attributed to Methylophilaceae, a bacterial family important for microbial carbon cycling in terrestrial and marine habitats [87]. The Methylophilaceae family has been explicitly linked to methanol dehydrogenase [88], one of the highly abundant genes discovered in this dataset, which participates in methanol oxidation. Verrucomicrobiae and Opitutae, both from the phylum Verrucomicrobia, contributed to the community differences in the pine biofilms at both wooden shipwreck sites, with the latter also being observed in 16S data from pine biofilms [30]. Opitutae have been discovered in the rhizosphere soil associated with a species of pine [89], while Verrucomicrobia were first observed in marine biofilms on glass in the Antarctic [90] and have been associated with early marine biofilm colonization [91].

The *Wsp* operon, responsible for regulating biofilm formation through chemosensory signal transduction in bacteria, plays a critical role in the initiation of biofilm formation upon surface contact, and it has been extensively studied in *P. aeruginosa* [92,93]. A recent study found *Wsp* systems to be conserved and broadly distributed across bacteria [94], making it an important model system to study biofilm formation in different environments. Regardless of shipwreck site, the steel biofilms overall had higher abundances of biofilm formation genes than the pine and oak biofilms, which may suggest that steel substrates have a greater potential for microbial attachment and initiation of biofilm formation than wood substrates, at least in short-term studies. While this observation is likely due to material properties, biofilms on steel submerged in natural seawater can form quickly (<2 weeks) [95], and the taxonomic composition of steel biofilms in a closed system can change within 30 days [42]. The data in this study are supported by pairwise PERMANOVA tests between substrates, which revealed the most significant differences between steel and both wood types, and by the higher abundance of the gene complex *WspE-WspR* in the steel biofilms. The *WspE-WspR* region of the *Wsp* operon codes for a response regulator protein within two-component signaling transduction systems in bacteria [96], which have been studied for their importance in detecting and responding to environmental changes, biofilm formation, and cell motility [97]. A prior study also discovered the importance of another response regulator gene, involved in the environmental stress response, within steel biofilms [73]. In addition to *WspE-WspR*, three other chemotaxis genes had the highest abundance at the closest distance to *Anona* and then decreased in abundance with increasing distance from the shipwreck, possibly implicating a microbial surface response on the steel to the presence of a shipwreck on the seabed. Chemotaxis is critical in surface-attached bacteria, and genes that code for chemotaxis proteins have been shown to significantly contribute to initial biofilm formation [2]. Although a similar abundance pattern of these genes was observed in the steel biofilms at Site 15711, the abundance then increased at the farthest distance; this discrepancy between sites may be ascribed to the size of the structure, as Site 15711 is smaller than *Anona*. 

From the set of biofilm formation genes targeted in this dataset, the pine and oak biofilms also had a higher average gene abundance of the response regulator protein *WspE-WspR*. However, except for the chemotaxis gene *CheR*, a higher abundance was observed in the oak biofilms, which may be related to differences in surface properties between the wood types. In a comparative study of the interaction of microbial adhesion on six types of wood, oak demonstrated the highest potentiality for the adhesion of bacteria and subsequent biofilm growth [98].

Below the photic zone, chemolithoautotrophic metabolisms are important for the regeneration of inorganic nutrients [99,100]. This microbial contribution has been observed in pelagic waters [101], in coastal sediments [102], and, more recently, in marine biofilms [103]. Targeting sets of genes corresponding to microbial chemolithotrophic metabolisms in this study revealed high abundances of sulfur and sulfide oxidation genes on the steel substrates, likely connected to the higher relative abundance of Epsilonproteobacteria (*Sulfurimonas* and *Arcobacter*), as shown previously [104]. While Epsilonproteobacteria had a greater abundance at *Anona*, a steel-hulled shipwreck, a greater abundance of genes related to sulfur cycling occurred at Site 15711, a wooden-hulled shipwreck, which may suggest an effect of substrate availability, as all three substrates were deployed at Site 15711, and only steel substrates were deployed at *Anona*. While the iron cycling (oxidation and reduction) genes in the steel biofilms displayed nearly the same abundances across these two sites, iron oxidation genes were higher at *Anona* and likely related to the presence of Zetaproteobacteria, demonstrated in recent studies [105,106].

Other chemolithotrophic metabolisms found to be more important in the steel biofilms were hydrogen oxidation, arsenate reduction, and nitrogen cycling. Genes involved in both nitrate reduction (*napA* and *napB*) and nitrite reduction (*nirB* and *nirD*) were detected within the steel biofilms; this has been observed previously on steel in seawater [107] and in deep subsurface biofilms [108]. Nitrogen cycling on steel surfaces may also be linked to microbially influenced corrosion [109]. Recently, a comparative study of marine microbial biofilms and planktonic communities in the deep ocean suggested the coupling of hydrogen oxidation and sulfate reduction to acquire nutrients and energy for biofilm formation through electron transport [102]. Arsenic is a metalloid toxin present in low abundances in the marine environment; often exists in oxygenated water in the form of arsenate [110]; and is known to be associated with iron oxides [111], which form as steel alloys corrode in salt water. While the adsorption of arsenic by iron oxides may remove arsenic from the water column [112], the presence and activity of the *arsC* gene in microorganisms may potentially increase the toxicity of arsenic in the local environment [113]. The *arsC* gene codes for a protein to reduce arsenate to arsenite, part of a mechanism that microorganisms developed to resist and detoxify arsenic through the *ars* operon [109]. Regardless of site, the steel biofilms had a higher abundance of this gene than the wood biofilms, which may be attributed to differences in physical properties between steel and wood substrates, such as surface roughness, chemical composition, and surface free energy.

A greater genetic potential for manganese oxidation, methanol oxidation, and urea hydrolysis was discovered in the pine and oak biofilms. While urea is produced by planktonic bacteria in the euphotic zone [114], the hydrolysis of urea produces ammonia and carbon dioxide, substrates used by ammonia-oxidizing bacteria for chemolithotrophic growth [115]. Urease genes have also been detected in archaea in the Arctic Ocean [116] and in the deep Mediterranean Sea [117]. The higher abundance of these genes within the wood biofilms suggests a greater potential of wood substrates for harboring microorganisms capable of metabolizing urea, which may potentially contribute to nitrogen cycling and regeneration in the deep ocean. Manganese-oxidizing microorganisms have been associated with manganese oxides, which commonly form on steel alloys [118]. While manganese oxidation genes, particularly the manganese oxidizing protein *MopA*, were found in biofilms on all substrates in this study, the abundance of this gene was the highest in the wood biofilms, which may suggest a connection to manganese-oxidizing bacteria, which oxidize manganese during lignin degradation [119]. C1 compounds contain no carbon–carbon bonds, and bacteria utilize these to obtain carbon and energy [120]. Genes related to C1 compounds were detected in all biofilms; however, this metabolism was dominated by a high abundance of a methanol dehydrogenase gene in the wood biofilms. The bacterial degradation of lignin yields methanol [121], used by methylotrophic bacteria, and potentially sulfate-reducing bacteria, as a carbon source for growth [75]. This metabolism contributes to carbon cycling in the marine environment [122], and it is likely associated with the abundance of Methylophilaceae (Betaproteobacteria) in biofilms. While methanol dehydrogenase was higher in abundance within the pine and oak biofilms, this gene was also found in the steel biofilms, with a higher abundance at Site 15711 than at *Anona*; this may also suggest an impact of substrate availability, or potential syntrophy between bacterial species within wood and steel biofilms, as these substrates were present in the same physical space on the seabed.

The marine biofilm microbiomes shared some commonalities regardless of substrate, such as biofilm-forming microorganisms and a few chemotaxis and two-component signaling genes, which were shared between sites. The taxonomic composition, however, of the biofilms diverged between the steel, pine, and oak substrates. Likewise, these substrates selected for gene abundances related to different chemolithotrophic metabolisms targeted in this study, including sulfur, iron, and nitrogen cycling, and hydrogen oxidation on the steel surfaces, and manganese and methanol oxidation on the pine and oak surfaces. Substrate primarily shaped biofilm microbiomes, specifically taxonomy, on new surfaces deployed to the seabed, and substrate availability may have contributed to the differences in metabolisms within the steel and wood biofilms across sites.

## 5. Conclusions

Anthropogenic expansion of the built environment, from coastal areas to the deep ocean, is anticipated to increase by at least 23% globally in the next 10 years [123]. Considering how submerged structures provide habitats for a diversity of marine life, the continuous addition of structures on the seabed raises new questions about their effects on biodiversity and, consequently, ecosystem function. Analyzing the functional potential of marine microbial biofilms in relation to different substrates on the seabed is a new step to understanding the consequences of anthropogenic activity in marine ecosystems, specifically on the seabed. The results of this study confirm prior findings that substrate type is a determining factor in recruitment to and activity within developing marine biofilms. As the built environment expands, it is important to document the changing landscape of the seabed to understand the effects on deep-sea ecology before more structures arrive.

## Figures and Tables

**Figure 1 microorganisms-11-02416-f001:**
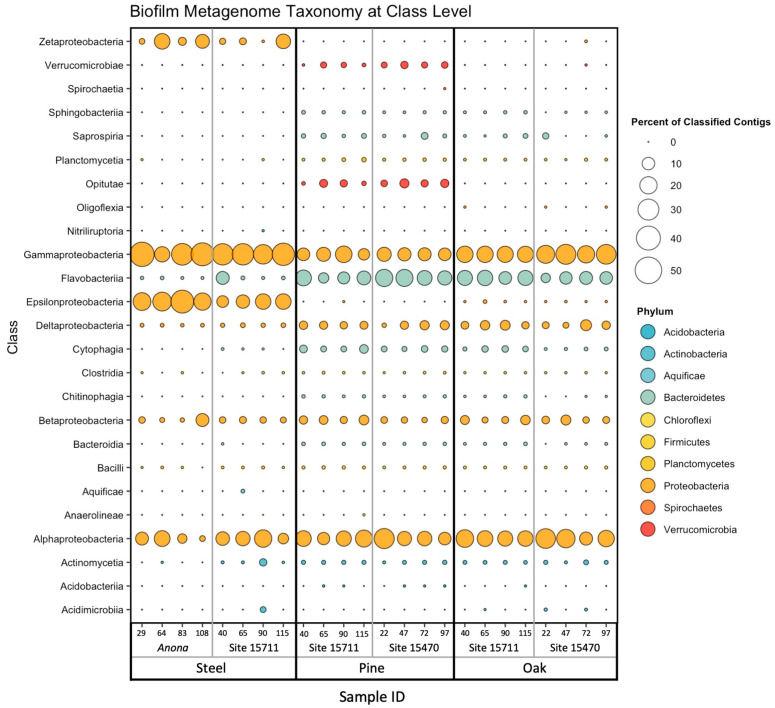
Bubble plot of relative abundance of biofilm metagenome taxonomy at the class level. Color corresponds to phylum, and size corresponds to percentage of classified contigs. Samples are arranged on the *X*-axis primarily by substrate, then by site, then by distance from structure in meters. Taxa composing <0.1% of community in each sample were excluded prior to constructing plot.

**Figure 2 microorganisms-11-02416-f002:**
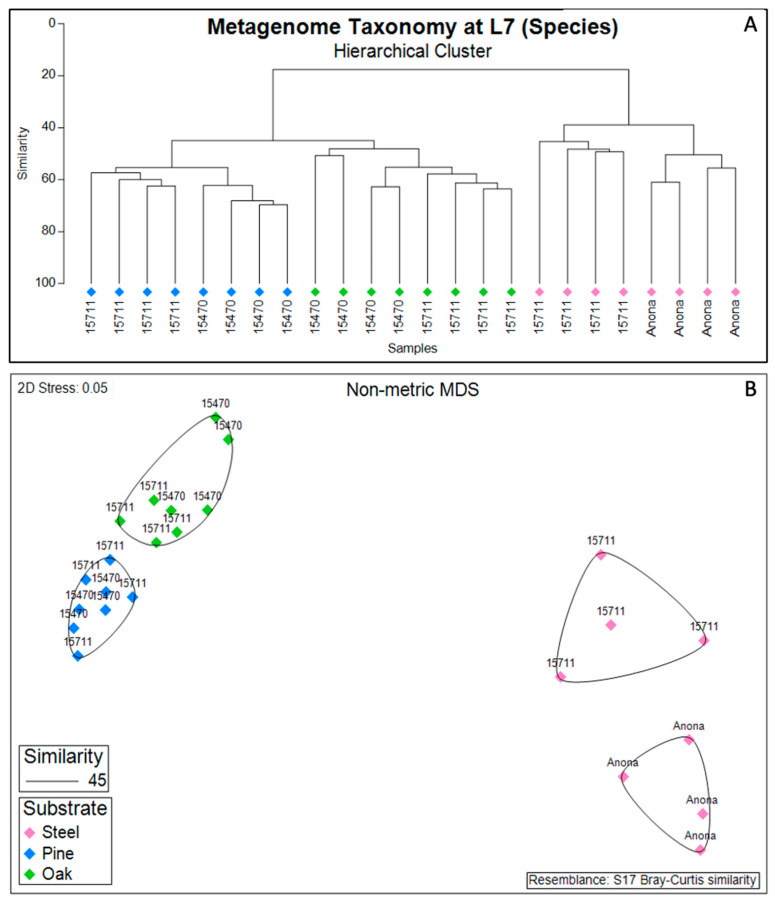
Hierarchical cluster analysis (**A**) of steel and wood biofilm metagenome taxonomy relative abundance based on a Bray–Curtis dissimilarity matrix and projected onto a non-metric multidimensional scaling (NMDS) plot (**B**). Data include all samples in dataset (*n* = 24). Icons are colored by substrate and annotated with site name; *Anona* is a historic, steel-hulled shipwreck, and Site 15711 and Site 15470 are historic, wooden-hulled shipwrecks.

**Figure 3 microorganisms-11-02416-f003:**
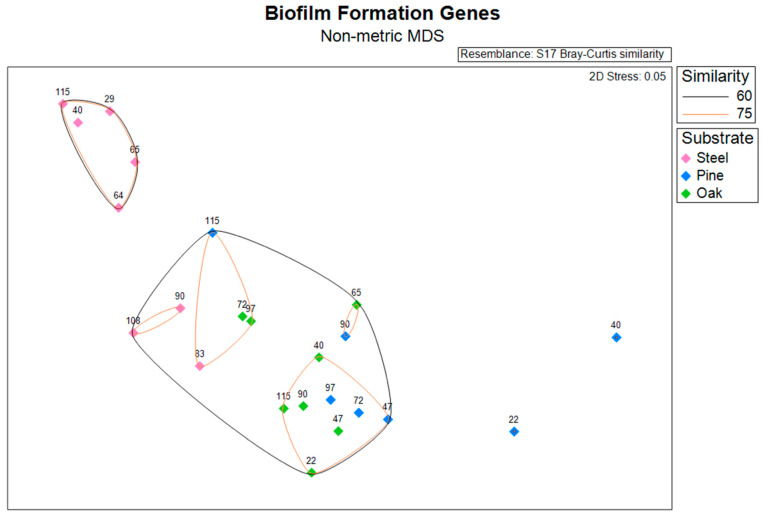
Non-metric multidimensional scaling (NMDS) plot based on a Bray–Curtis dissimilarity matrix of normalized gene abundance for biofilm formation genes. Icons are colored by substrate and annotated with distance from structure in meters.

**Figure 4 microorganisms-11-02416-f004:**
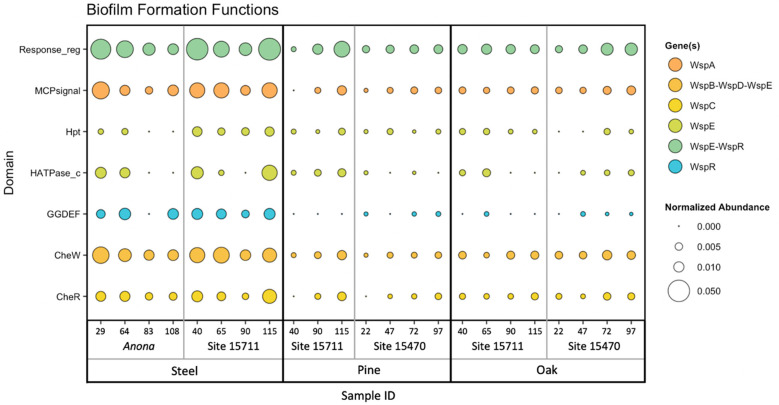
Bubble plot of normalized gene abundance of a set of biofilm formation genes within metagenomes. Color corresponds to gene name, and size corresponds to normalized abundance. Samples are arranged on the *X*-axis primarily by substrate, then by site, then by distance from structure in meters.

**Figure 5 microorganisms-11-02416-f005:**
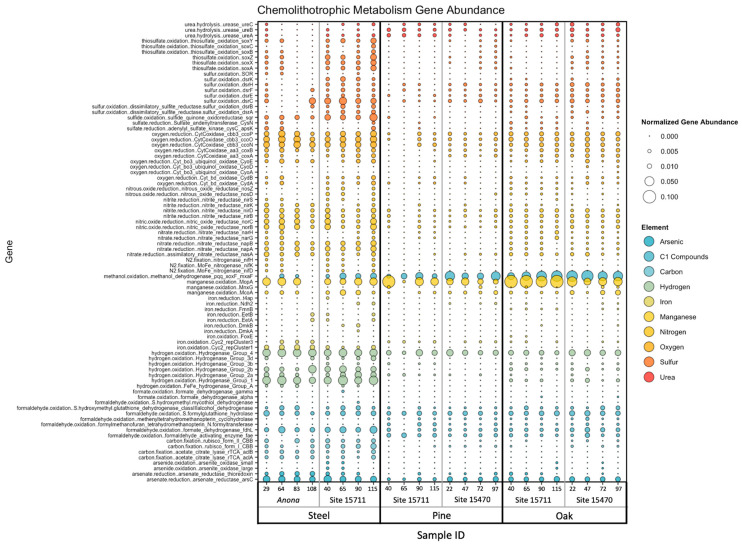
Bubble plot of normalized gene abundance of individual genes related to microbial chemolithotrophic metabolisms within metagenomes. Color corresponds to element, and size corresponds to normalized abundance. Samples are arranged on the *X*-axis primarily by substrate, then by site, then by distance from structure in meters.

## Data Availability

Metagenome sequences were submitted to NCBI under BioProject PRJNA949243.

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
