# Peer review of "Substrate Specificity of Biofilms Proximate to Historic Shipwrecks"

_microorganisms, 2023, doi:10.3390/microorganisms11102416_

Round 1
Reviewer 1 Report
Review by article microorganisms-2562958
The article "Substrate specificity of biofilms proximate to historic shipwrecks" by Rachel L. Mugge, Rachel D. Moseley and Leila J. Hamdan studies the microbial communities of biofilms formed on different substrates (steel, wood) in the marine environment. The authors attempt to assess the influence of anthropogenic factors on the formation of microbial biofilm communities. Undoubtedly, this study is in demand from the point of view of ecological monitoring of marine areas. A major comment on the publication is the lack of information on fungal communities in both the results and discussion sections. It is recommended that the fungal community data either be removed or significantly expanded. The majority of the paper to date has focused on bacteria and bacterial genes.
Abstract
This section fully captures the essence of the study and is written according to journal requirements.
1. Introduction
This section should add information, based on literature data, on which bacterial taxa are prevalent on wood and steel surfaces in marine biofilms.
2. Methods
11) 2.1 Field experiments
Can we compare the microbial communities of biofilms from man-made metal and wood structures near ships at different depths?
2) The remaining subsections of 2. Methods are very detailed. However, it is not clear - did the authors study only the bacterial community or both bacterial and fungal communities?
3. Results
In section 3.3. Taxonomic composition of the biofilm, there is no characterisation of the fungal taxa, although fungi are listed in Figure 1. The authors should either add a characterisation of the fungal community taxa or remove them from the figure.
4. Discussion
1) Lines 515-517 - " In addition to the eight fungal classes detected in 16S analysis of pine and oak biofilms [24], the fungal class Basidiomycota was also abundant (>1%) within metagenome taxonomic profiles for both wood types." Fungi are detected by ITS sequence analysis, not 16S. Why do the authors quote such a statement? The reference cited states: " For fungal analysis, amplification and sequencing of the internal transcribed spacer region 2 (ITS2) was performed at the Integrated Microbiome Resource facility with primers ITS86F/ITS4, which have been shown to be suitable for studying diversity and community structures of fungi (Op de Beeck et al., 2014 ).".
2) Lines 540-550 are recommended to be moved to a separate conclusion section.
References
Out of 116 references, only 30 references are newer than 2018, it is advisable to add information on specific taxa in the introduction and thus increase the number of references to publications from the last five years. Self citations - 9 references - 17, 24, 34, 35, 35, 36, 37, 38, 39, 67. It is recommended to include the doi of articles in the references.
Line 700 - Reference 62 - year should be bold
Line 684 - Reference 54 - year should be bold
Line 656 - reference 39 - has the article not yet been published? Is it correct to refer to it?

Minor editing of English language required
Reviewer 2 Report
The authors present a study evaluating the colonization of bacterial communities on different substrates that may be components of shipwrecks and/or infrastructure. The study adds to the current literature in this area and adds unique components of distance and multiple sites with metagenomic analyses of these communities which goes beyond the traditional amplicon sequencing. This study helps us better understand the functional potential of the colonizing biofilm communities on various substrates. I also commend the authors on a well-written paper which is often difficult with this type of data. Some additional justification for the experimental design is warranted to help interpret the results and some additional analyses would better leverage the metagenomic data and their overall findings. See below for recommendations and comments.
The experimental design includes deploying three substrate coupons (steel, pine, and oak) near a steel-hulled wreck as well as two wooden-hulled wrecks. However, only at Site 15711 were all three substrate types deployed. Why weren’t all 3 types deployed at all 3 wreck sites?
Similarly, as we know that environmental location and the presence of the wreck itself will impact the colonization of these biofilm communities, it’s hard to separate the impact of the wreck on the biofilm communities and the substrates themselves. What is the colonization of these substrates near other wrecks representing in the larger picture of how microbes colonize new anthropogenic sources on the seabed? Is the thought that they’d be entering the seabed near other structures?
Also, it’s hard to assess how different the two factors “distance” and “site” are without information about how far the wrecks are from each other. The authors address this a bit in the results and discussion, but it warrants further clarification in the methodology and experimental design to help interpret and analyze the results. To also address this further, it may be beneficial to focus more of the results from the Site 15711 where the impact of the wreck and location are all relatively equal and therefore, will allow you to compare substrate-specific effects.
Methods 2.1- Can you include how far apart these wrecks are from each other as well? This will help the reader better understand how far apart the distance is between the sites (see comment above). Please also include how the distances from the wreck and deployment time were chosen. The reasoning behind these decisions is largely unclear.
The inclusion of the metagenomic data is a strength of this study and is largely used to look at taxonomic abundances as well as general gene abundances of the whole communities. However, this data can also be leveraged to look at who the genomic potential belongs to. This would better assess the specific role of these taxa and how these findings have advanced our understanding of colonization of these substrates, better linking the two datasets and filling in some subsequent questions about interactions. This can be done by analyzing the functional genes in a program last BLAST and/or developing MAGs for some of the most abundant taxa.
The finding that the biofilm-related genes are influenced by distance from the wreck for both steel and wood (though different effects) is extremely interesting. This suggests a wreck-specific effect and has implications for the presence of established structures around a new one and how it could influence colonization. This may be something worth expanding on in the discussion section and putting it into context of the previous studies that have found similar things (which the authors mention briefly). Or if you think there are other things influencing these results, please expand on those.
Figure 2. Can you modify this figure to show which samples are from which distance from the wreck? This will provide readers with the most data for interpretation of the results. For example, in instances for oak samples where 15470 and 15711 samples are closer to each other than the other samples from their site, does distance play a factor at all? While this is discussed a bit in the results, it would be good to see this accounted for in the figure itself.
Consider moving some of the diversity data (Supplemental Tables S1-S3) to the main text. These results are interesting of themselves and warrant additional discussion.
Discussion Line 384- rephrase this sentence as it suggests that you analyzed colonization of both steel and wood at Sites 15711 and 15470, but I don’t see the data for steel at 15470.
Reviewer 3 Report
I think that this paper is an important contribution to biofilm research. The English is perfect. It should be published.
I have only a few comments, as follows:
lines 219-230 I don't think this level of detail is required. Is it important for understanding the results and discussion? Could it be removed?
In lines 295-6, remove (wooden hulled) and instead simply use pine and oak respectively.
Explaining all the methods in this level of detail is tiring for the reader. I wonder if much would be lost if you omitted "distance" from this paper and used it for another publication - after all, it makes little (though a bit of) difference. Just a suggestion.
line 396 Which wood type was rougher? If you don't have this information, don't mention roughness.
line 501 Which surface property could have this effect?
Round 2
Reviewer 2 Report
The authors have responded clearly and thoroughly to the reviewer comments. The additional clarification on the experimental design, statistical analyses, and context in the overall literature have improved the clarity of the study and interpretation of the results. While some of the recommended changes would have expanded the impact of the study and its findings, their rationale for it being outside the scope of their objectives is reasonable.